# Conformational plasticity and dynamic interactions of the N-terminal domain of the chemokine receptor CXCR1

**Shalmali Kharche**[1,2], **Manali Joshi**[3], **Amitabha Chattopadhyay**[4]*,
**Durba Sengupta**[1,2]*

**1** CSIR-National Chemical Laboratory, Pune, India, **2** Academy of Scientific and Innovative Research (AcSIR), Ghaziabad, India, **3** Bioinformatics Centre, S. P. Pune University, Pune, India, **4** CSIR-Centre for Cellular and Molecular Biology, Hyderabad, India

* amit@ccmb.res.in (AC); d.sengupta@ncl.res.in (DS)

## Abstract

The dynamic interactions between G protein-coupled receptors (GPCRs) and their cognate protein partners are central to several cell signaling pathways. For example, the association of CXC chemokine receptor 1 (CXCR1) with its cognate chemokine, interleukin-8 (IL8 or CXCL8) initiates pathways leading to neutrophil-mediated immune responses. The N-terminal domain of chemokine receptors confers ligand selectivity, but unfortunately the conformational dynamics of this intrinsically disordered region remains unresolved. In this work, we have explored the interaction of CXCR1 with IL8 by microsecond time scale coarse-grain simulations, complemented by atomistic models and NMR chemical shift predictions. We show that the conformational plasticity of the *apo*-receptor N-terminal domain is restricted upon ligand binding, driving it to an open C-shaped conformation. Importantly, we corroborated the dynamic complex sampled in our simulations against chemical shift perturbations reported by previous NMR studies and show that the trends are similar. Our results indicate that chemical shift perturbation is often not a reporter of residue contacts in such dynamic associations. We believe our results represent a step forward in devising a strategy to understand intrinsically disordered regions in GPCRs and how they acquire functionally important conformational ensembles in dynamic protein-protein interfaces.

## Author summary

How cells communicate with the outside environment is intricately controlled and regulated by a large family of receptors on the cell membrane (G protein-coupled receptors or GPCRs) that respond to external signals (termed ligands). Chemokine receptors belong to this GPCR family and regulate immune responses. We analyze here the first step of binding of a representative chemokine receptor (CXCR1) with its natural ligand, interleukin-8 (IL8) by an extensive set of molecular dynamics simulations. Our work complements previous mutational and NMR experiments which lack molecular-level resolution. We show that in the inactive state, one of the extracellular domains of the CXCR1 receptor, namely

**Data Availability Statement:** All simulations and analysis have been performed with open source tools such as GROMACS simulation package and utility tools. The starting files for the simulations

and some of the output files (i.e., the binding modes) have been uploaded as compressed Supporting Information files.

**Funding:** This work was supported by the Science and Engineering Research Board (Govt. of India) project (EMR/2016/002294) to D.S. and A.C. A.C. gratefully acknowledges support from SERB Distinguished Fellowship (Department of Science and Technology, Govt. of India). S.K. thanks the Council of Scientific and Industrial Research, Govt. of India, for the award of a Senior Research Fellowship. The funders had no role in study design, data collection and analysis, decision to publish, or preparation of the manuscript.

**Competing interests:** The authors have declared that no competing interests exist.

the N-terminal domain, is highly flexible and like a "shape-shifter" can exist in multiple conformational states. However, when IL8 binds, the N-terminal domain undergoes a conformational freezing, and acquires a C-shaped "claw-like" structure. The complex between the receptor and IL8 is still quite dynamic as this C-shaped N-terminal domain forms an extensive but slippery interface with the ligand. We further corroborated these results by quantitative comparison with NMR and mutagenesis studies. Our work helps clarify the inherent disorder in N-terminal domains of chemokine receptors and demonstrates how this domain can acquire functionally important conformational states in dynamic protein-protein interfaces.

## Introduction

G protein-coupled receptors (GPCRs) are an important class of membrane-embedded receptors that respond to a diverse range of stimuli [1,2]. These receptors play a central role in several cellular signaling pathways, and consequently are targeted by a large number of drugs [3,4]. Recent advances in GPCR structural biology have helped to resolve the structure of transmembrane domains of several GPCRs. However, the interconnecting loops and the N- and C-terminal extramembranous regions remain largely unresolved [5,6]. The high flexibility of these loop regions makes it challenging to resolve their conformational states, but at the same time gives them a functional significance [6,7]. Both direct interaction (*e.g.*, between intracellular loop 3 (ICL3) and effectors [8]) and allosteric modulation by extramembranous loops (such as extracellular loops 2 and 3 (ECL2, ECL3) [6,9,10]) have been reported in various GPCRs. The N-terminal region, known to interact with ligands [11] in GPCRs such as chemokine receptors [12–14], is of special interest in this context. In addition, N-terminal population variants of several GPCRs have been reported to alter drug response by allosteric modulation of ligand binding [15–17]. Interestingly, lipid specificity and conformational sensitivity of extramembranous regions in GPCRs have recently been reported [18–20]. In spite of their functional role, extramembranous regions in GPCRs remain largely uncharacterized in terms of their structure and dynamics.

Chemokine receptors are members of the GPCR superfamily that bind chemokine secretory proteins and play a fundamental role in innate immunity and host defense [21,22]. These receptors highlight the functional importance of the N-terminal region since it represents the first site of ligand binding and confers selectivity to these receptors [23]. A common two-site/two-step model has been proposed for chemokine binding that suggests interactions between receptor N-terminal domain and chemokine core (chemokine recognition site-I, CRS1) and between the chemokine N-terminus and receptor extracellular regions or transmembrane residues (site-II) [23–25]. In addition, recent reports confirm that the stoichiometry of binding is 1:1, although both the receptor and chemokines have been shown to dimerize in the cellular milieu [24–26]. Early attempts to structurally characterize these complexes focused on site-I interactions and solution nuclear magnetic resonance (NMR) approaches were successful in resolving the interactions between chemokines and short receptor fragments without the context of the full-length receptor or membrane environment [27,28]. More recently, crystal structures have resolved site-II interactions, but only a partial site-I engagement [29,30]. However, a superposition of structures with respect to the bound chemokine indicates that the placement of the receptor N-terminus could be receptor-specific [31]. Although the two-site model served as the initial framework of functionally relevant interactions leading to chemokine-receptor binding, growing literature suggests a need for more complex models accounting for the dynamic mechanism of receptor-ligand binding [32].

The CXC chemokine receptor-1 (CXCR1) is a representative chemokine receptor that controls the migration of neutrophils to infected tissues [33]. The three-dimensional structure of CXCR1 (residues 29–324) has been elucidated by solid state NMR [34] and follows a typical GPCR fold, with seven transmembrane α-helices interconnected by three intracellular and three extracellular loops. The two flanking domains, the extracellular N-terminal and intracellular C-terminal regions, were not resolved in this structure. CXCR1 binds the CXC ligand, CXCL8, commonly termed interleukin-8 (IL8). There are several reported structures of IL8 in monomeric and dimeric forms, although none bound to CXCR1 [27,28,35]. Several studies have highlighted a crucial role of the N-terminal region of CXCR1 in ligand binding affinity and selectivity [36]. The interactions of IL8 were assessed using NMR with CXCR1 constructs of varying length, clearly indicating that IL8 could not bind to CXCR1 when the receptor N-terminal was truncated [37]. In addition, IL8 was shown to bind with higher affinity to the CXCR1 N-terminal region in a lipid environment relative to that in solution [36], in agreement with our previous work using fluorescence and molecular dynamics (MD) simulations which show membrane interaction of the CXCR1 N-terminal region [38–40].

In this work, we have examined chemokine-receptor interaction focusing on the N-terminal region of CXCR1 and its role in chemokine binding. An overview of the receptor embedded in the membrane and its structural domains is shown in Figs 1 and S1, respectively. We performed simulations of *apo*-CXCR1 as well as CXCR1 coupled with IL8 at coarse-grain and atomistic resolutions to monitor differential dynamics of the N-terminal region. We show that the N-terminal region is the first site of chemokine binding which restricts its conformational dynamics. The receptor-chemokine (CXCR1-IL8) complex consists of an extensive dynamic interface and we map the interactions both within the receptor and with the ligand. These results were further validated by comparison with chemical shift calculations reported in earlier NMR studies. Our results offer molecular insight into the interactions between CXCR1 and IL8, and would be useful in gaining a fundamental understanding of the initial events in chemokine-receptor interactions at site-I (CRS1).

## Results

The N-terminal region of the chemokine receptor CXCR1 remains structurally unresolved in experiments due to its inherent flexibility [34]. The importance of this region is reflected in reports that implicate it in the binding of the cognate chemokine (IL8) [36,37], similar to all members of the chemokine receptor family [14]. To explore the underlying molecular interactions, we have performed coarse-grain molecular dynamics simulations of CXCR1 and corroborated them against atomistic models. An overview of the N-terminal region and the other structural domains of CXCR1 and IL8 is provided in S1 Fig. We report here the functional dynamics of the N-terminal region of CXCR1 in the *apo-* and IL8-bound forms.

### Conformational plasticity of the N-terminal region of *apo*-CXCR1

Coarse-grain simulations of the *apo*-CXCR1 receptor were performed starting from the extended N-terminal conformer (Fig 1A). In total, twenty simulations were performed totaling to 200 μs. During the simulations, the N-terminal region diverged from the initial structure and appeared to be quite dynamic. The N-terminal region sampled several orientations and was found to interact at different time points with the membrane bilayer and the transmembrane domains. The two main orientations observed (membrane-bound and receptor-contacted conformers) are shown in Fig 1B and 1C. These can be distinguished by the distance of distal residues 1–10 of the N-terminal region from the membrane (see Fig 1D). In the initial placement, the N-terminal region is far from the membrane (yellow stretches in the plot), and

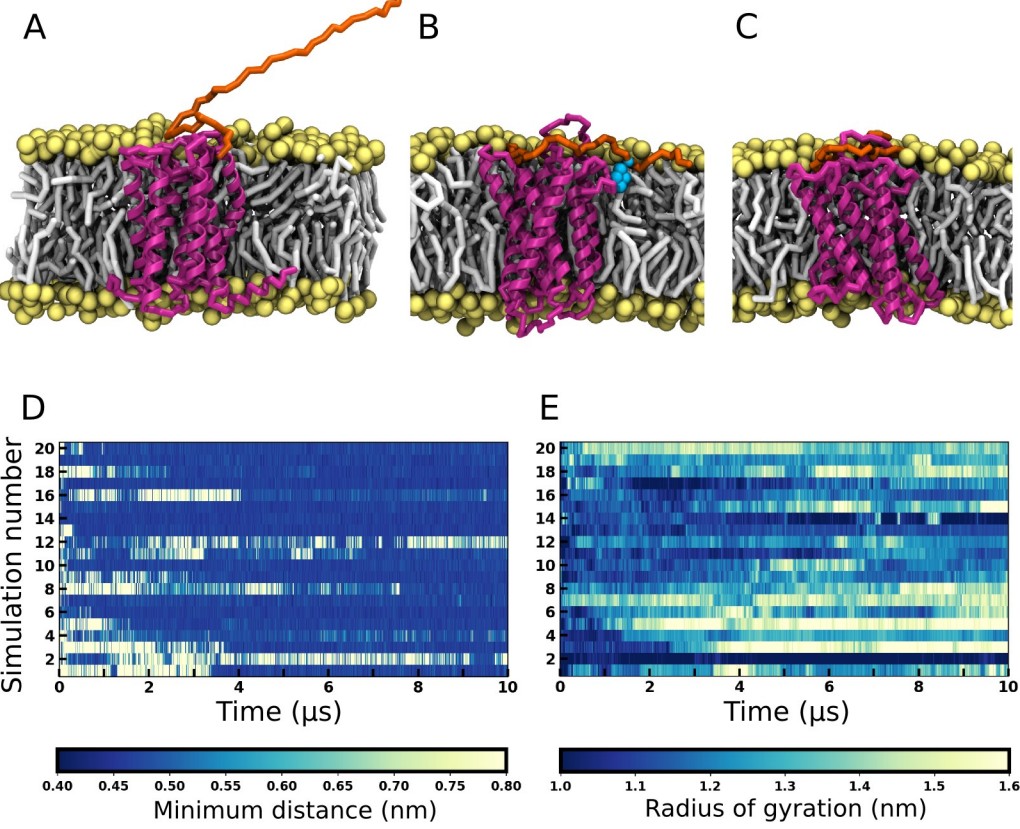

**Fig 1. Representative snapshots of CXCR1 embedded in a lipid bilayer and membrane interaction of its N-terminal region.** A visual representation of (A) the starting conformation with an extended N-terminal region, (B) the membrane-embedded N-terminal conformer and (C) the receptor-contacted N-terminal conformer. The receptor is depicted in magenta, the N-terminal region in orange, and the lipid headgroups and tails in yellow and gray, respectively. Water and ions are not displayed for clarity. The residue W10 of the N-terminal region, which interacts with the lipid bilayer is shown as cyan colored beads. (D) The minimum distance between the lipid bilayer and the distal part of the N-terminal region (residues 1–10) is plotted for 20 simulations of *apo*-CXCR1 as a function of time. The color bar denotes minimum distance in nm. A distance of ~0.4 nm (dark blue patches) indicates the binding of the N-terminal region to the lipid bilayer. (E) The radius of gyration of the N-terminal region is plotted for *apo*-CXCR1 as a function of time. The color bar denotes radius of gyration in nm. See Methods for more details.

relaxes in a nanosecond timescale to interact with the membrane (blue stretches). Several close interactions with the membrane (blue stretches) and multiple association-dissociation events were observed (see Fig 1D). When the N-terminal region dissociated from the membrane, it was located on top of the receptor, interacting with the transmembrane helices. In this orientational state, it appeared to be more compact, as reflected in the radius of gyration (see Fig 1E). Overall, the orientation and position of the N-terminal region in the *apo*-receptor was highly dynamic.

To test the conformational landscape sampled in the coarse-grain simulations, we performed all-atom simulations of CXCR1 embedded in the membrane bilayer (see S2 Fig). The N-terminal region of CXCR1 adopted multiple conformations, and no stable secondary structure was observed over time (S2B Fig). For a direct comparison, the intra-protein contacts were computed from both coarse-grain and atomistic simulations. Several off-diagonal elements were observed in both cases representing close interactions between residues which are sequentially apart (S2A Fig). The off-diagonal contacts in the middle of the N-terminal region (around residues 20–25) indicate a compact conformation. Interestingly, we observed similar

patterns in the contact maps (S2 Fig), indicating that the coarse-grain simulations were able to capture the overall conformational dynamics of this highly flexible region.

## The N-terminal region is the first site of ligand binding

We carried out coarse-grain simulations of CXCR1 with IL8 to examine the effect of ligand binding upon the structural dynamics of the N-terminal region of CXCR1. Overall, forty simulations were performed with two conformations of CXCR1 N-terminal region (membrane-bound and receptor-contacted) and two placements of IL8 (N-domain of the ligand facing the receptor and away from it). During the course of the simulations, IL8 diffused randomly in water and was observed to bind to the membrane-embedded CXCR1 within microseconds. A representative snapshot of the CXCR1-IL8 complex is shown in Fig 2A. The binding events were quantified from the minimum distance between IL8 and the receptor (Figs 2B and S3). The distance around 0.5 nm (blue stretches in the plot) indicate close interactions between the two proteins. A few binding-unbinding events were observed before the CRS1 bound complex was formed and no further unbinding was observed during the course of the simulations.

To understand the mechanism of binding, we characterized the interaction between the receptor domains and IL8 from the time of binding (Fig 2C). The time point corresponding to the binding event (time of binding t = 0) is considered to be the time frame where the CRS1 bound complex is formed (taken from Fig 2B). For clarity, the receptor domains considered were the N-terminal region, the three extracellular loops (ECL1-3) and the lumen defined as the residues from the transmembrane helices lining the top of the receptor lumen. The minimum distance (distance of closest contact) between these domains and IL8 was calculated from the time of binding and averaged over all simulations. Interestingly, the N-terminal region was observed to be the first site involved in binding of IL8 (Fig 2C). Subsequently, IL8 was observed to interact with ECL3 followed by ECL2 and the lumen, and ECL1 does not appear to make any contacts. These contacts are maintained till the end of the simulations (10 μs after the initial binding) and the interactions with the N-terminal region appear quite stable. No interactions were observed with ECL1 consistent with the initial binding mode. We observed that the interactions with ECL3 reduced and that with ECL2 and the top of the lumen increased with time. We were unable to discern a deeper binding of the N-domain of IL8 in the receptor lumen. Overall, we observed that the N-terminal region of CXCR1 is the first site of binding for IL8 and this contact is maintained throughout the course of the simulations along with additional contact with sites on ECL2, ECL3 and the lumen.

## Conformational restriction in the N-terminal region upon ligand binding

To analyze the effect of ligand binding on conformational dynamics of the N-terminal domain, we computed intra-protein contact maps of the N-terminal region in the ligand-bound complex. These contact maps represent pair-wise probabilities of interaction for each residue pair within the N-terminal region, averaged over simulation time and all simulation sets. A composite contact probability map displaying direct comparison of residue-wise contacts within the N-terminal region from *apo*-CXCR1 (upper diagonal) and CXCR1-IL8 complex (lower diagonal) is shown in Fig 3. Interestingly, several intra-protein contacts observed in the *apo*-receptor appear to be lost in the ligand-receptor complex and the N-terminal region appears to be more open in the ligand-receptor complex. A few intra-protein contacts were observed in the distal region of the N-terminal region in the ligand-bound complex, but appear to be relatively weak. We identified a few representative inter-residue contacts that dynamically form in the *apo*-receptor, but are completely absent in the ligand-bound simulations (S4 Fig). These

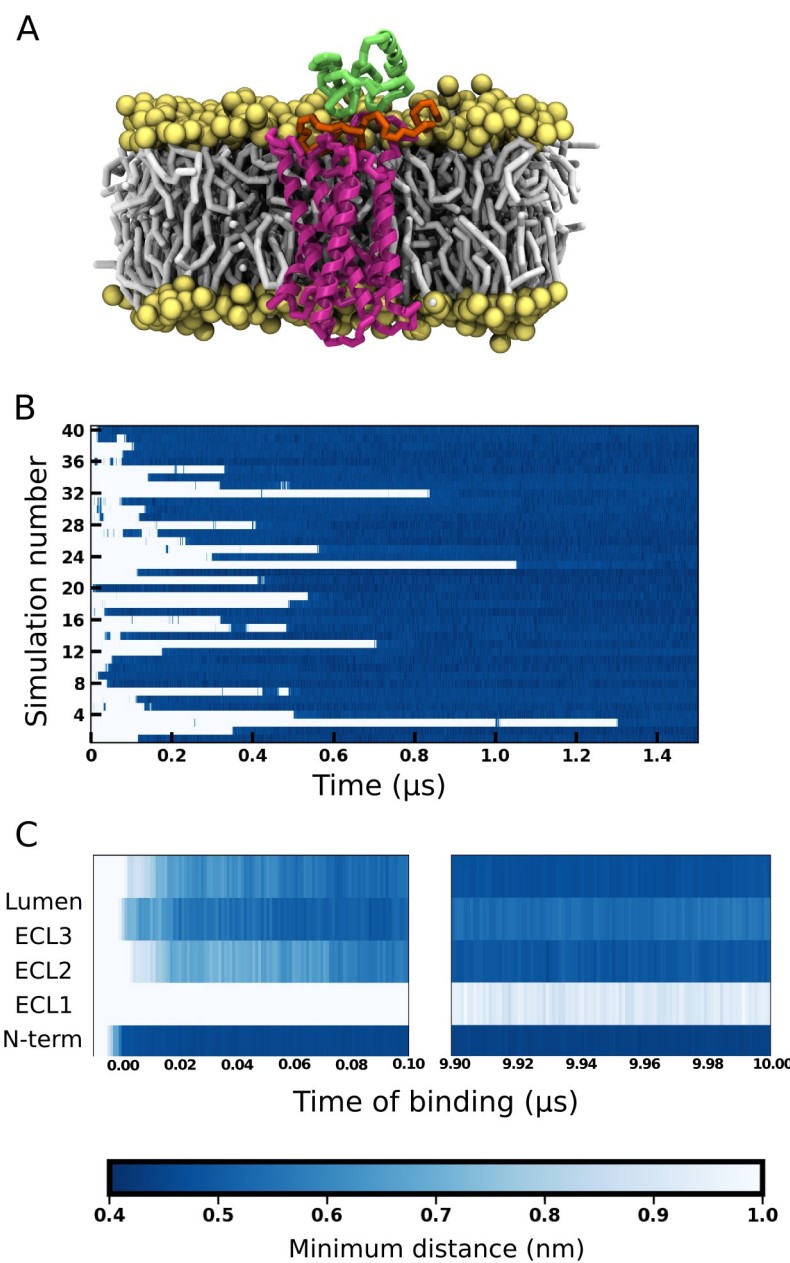

**Fig 2. Interactions between the extracellular domains of CXCR1 and IL8.** (A) A representative snapshot of IL8 bound to CXCR1. The receptor is shown in magenta, IL8 in green, and lipid headgroups and tails in yellow and gray, respectively. The N-terminal region of the receptor is highlighted in orange. Water molecules and ions are not shown for clarity. (B) The minimum distance (closest approach) between IL8 and CXCR1 plotted for the first 1.5 μs in forty simulations. The white stretches represent the unbound regime and the blue stretches represent the ligand-bound regime. Time of binding (t = 0) is defined as the time of first contact in the binding regime (0.5 nm distance cutoff) which remains undissociated till the end of the simulation. (C) The minimum distance between IL8 and various domains of the receptor as a function of time, considering the time of binding as t = 0. The values are averaged over all sets from the time of binding and plotted for the first 100 ns (left panel) and the last 100 ns (right panel). The color bar denotes minimum distance between IL8 and CXCR1 domains. See Methods for more details.

interactions include electrostatic interaction (Met1-Asp26), putative hydrogen bonding (Thr5-Thr18, Ser2-Thr18) and aromatic ring stacking (Phe17-Tyr27).

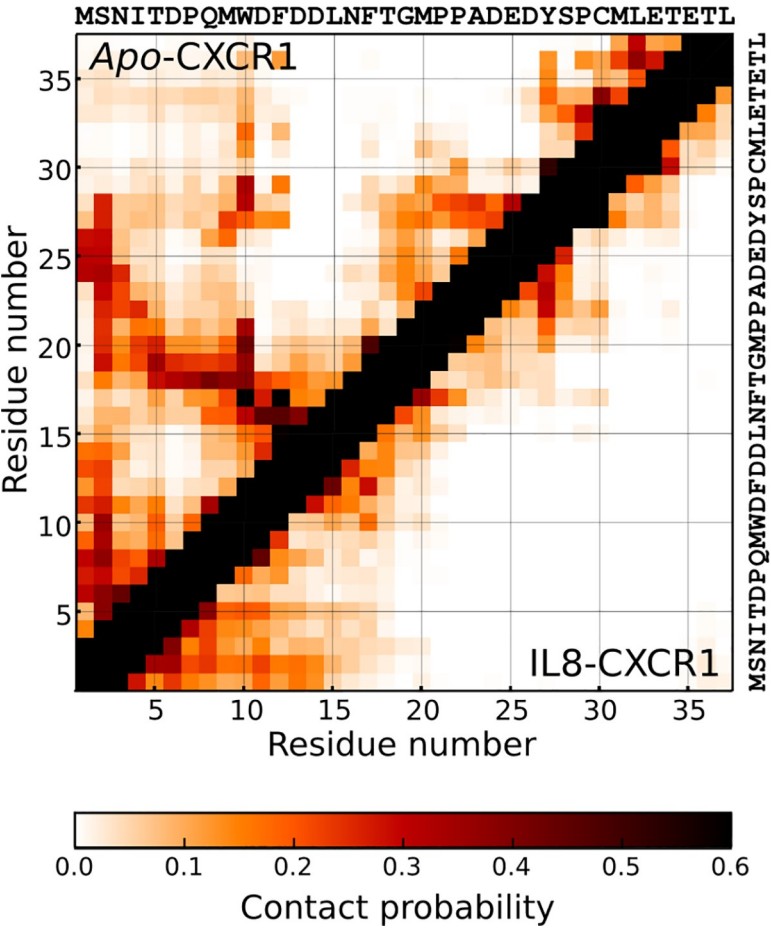

**Fig 3. Conformational dynamics of the N-terminal region of CXCR1.** Intra-protein contact maps of the N-terminal region of CXCR1 in presence (lower matrix) and absence (upper matrix) of the ligand. Residue-wise contact probabilities of the N-terminal region in *apo-* and IL8-bound CXCR1 are plotted in the top and bottom diagonal of the matrix, respectively. The amino acid sequence of the N-terminal region is displayed on the top and right. The values of contact probabilities (0.5 nm distance cutoff) are denoted in the color bar. See Methods for more details.

A more detailed characterization of the conformational dynamics was carried out by projecting the simulation trajectories onto a two-dimensional phase space. The two collective variables considered for the projection were the backbone RMSD of the N-terminal region and the distance distribution of an inter-residue contact Met1-Asp26 (Fig 4). The backbone RMSD describes an overall structural deviation with respect to a reference structure corresponding to the highest population cluster. The second reaction coordinate, *i.e.*, the distance between N-terminal residues Met1 and Asp26, reports on the end-to-end distance of the N-terminal region. We then calculated the normalized population of the N-terminal region by projecting the phase space along these reaction coordinates. Fig 4 represents the relative populations sampled in all the *apo-* and IL8-bound CXCR1 simulations (binned and averaged over all simulation sets). Multiple clusters were observed in the conformational landscape of the *apo*-receptor (marked I-III in Fig 4A), but only a single broad cluster (I) was observed in the ligand-bound receptor. The major cluster (cluster I in Fig 4B) in the IL8-bound simulations consists of conformers with a high end-to-end distance but low RMSD. The main cluster (cluster II in Fig 4A) in the *apo*-receptor exhibits a high RMSD. Interestingly, cluster I in the *apo*-receptor appears to overlap with a part of the conformational space sampled in the ligand-bound

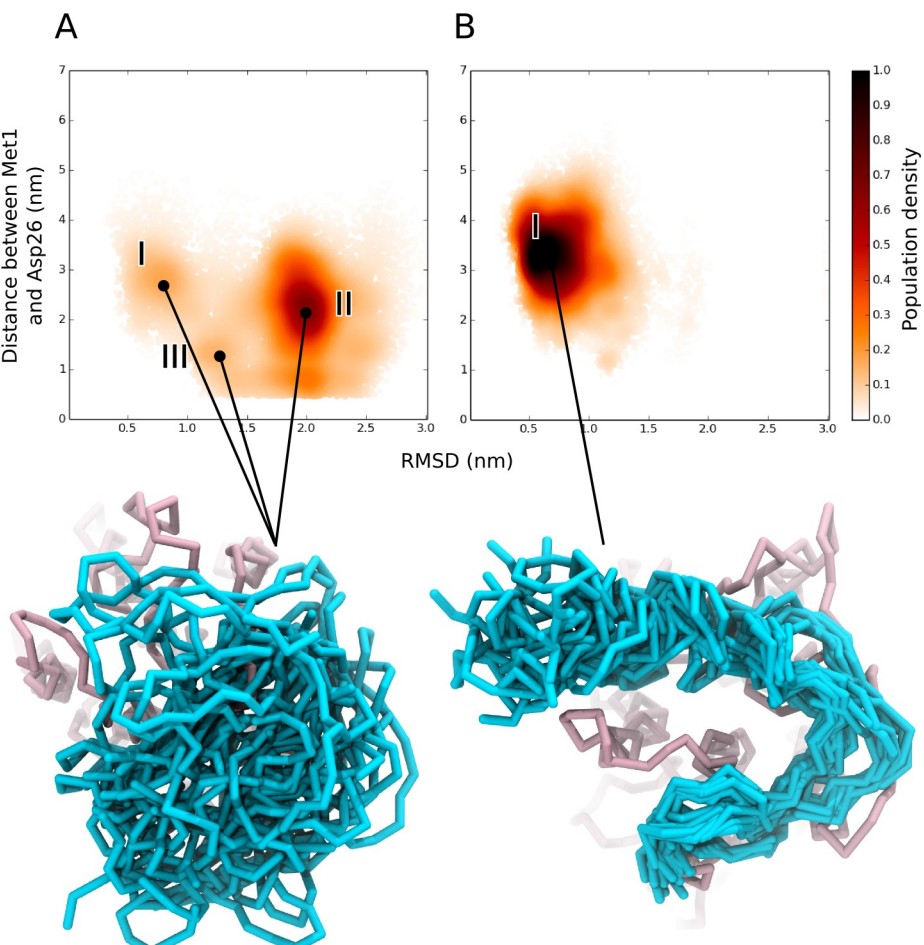

**Fig 4. Conformational landscape of the N-terminal region of CXCR1.** Population density map of the conformations sampled by the N-terminal region plotted as a function of backbone RMSD of the N-terminal region and the distance between side chains of two representative residues (Met1 and Asp26) for (A) *apo*-CXCR1 and (B) IL8-bound CXCR1. The most populated conformations are shown below the plots. The N-terminal region is shown in cyan and rest of the receptor is in pink. See Methods for more details.

complex. The main representative structures of these clusters are shown in Figs 4 and S5 and highlight the variable dynamics of the N-terminal region in the *apo*- and IL8-bound receptor.

The single cluster in the ligand-bound complex (Fig 4B) appears to be in contrast to the lack of intra-protein contacts observed in the receptor-ligand simulations (see Fig 3). A visual inspection revealed that the ligand-bound structures adopt a C-shape in the N-terminal region (Fig 4B). Such a conformation allows a more extensive protein-protein interface when the ligand is bound to the receptor, but at the same time results in the loss of intra-protein contacts. To characterize this C-shaped state, we calculated the contact maps of the interactions between the N-terminal region and the extracellular loops (S6 Fig). Interestingly, we observed large differences in the interactions in the *apo*- and IL8-bound N-terminal region. The N-terminal region of the *apo*-receptor samples several interaction sites on the extracellular loops and we could not discern a consensus pattern of interacting residues, confirming the presence of diverse conformational states. In contrast, specific regions of the N-terminal region were found to interact with each of the extracellular loops in case of IL8-bound receptor, giving rise to a C-like shape.

## Mapping the N-terminal region interactions: Corroboration by NMR chemical shift perturbations

We analyzed the molecular interactions of the N-terminal region by calculating the contact probabilities with the IL8 chemokine (see Fig 5A). We observed an extensive contact surface between the ligand and the N-terminal region, and a large number of flexible contacts were observed along the length of the N-terminal region. The contact map is consistent with the C-shaped N-terminal region described above with maximal contact probabilities at the center of the region. In particular, a high contact probability is observed at residues 20–25. The residues predicted to have a high contact probability match well with previous mutagenesis data. In particular, residues Pro21 and Tyr27 have been previously shown by mutational studies to be critical for ligand binding [41].

One of the few experimental approaches that are able to report conformational dynamics of this region is NMR using chemical shifts of the backbone amides that are closely related to their conformations. Chemical shift perturbations between the *apo-* and IL8-bound CXCR1 receptor from NMR studies in lipid environments have previously been reported [37,42]. To compare this data with simulations reported here, we chose representative structures from each of the coarse-grain simulation sets and mapped them to their atomistic representation. Subsequently, we computed the predicted chemical shifts in the backbone amides of N-terminal region using Eq (1). The resultant chemical shift perturbations plotted as a function of residue number are shown in Fig 5B. We observe that the central segment of the N-terminal region (residues 10–19) shows a higher chemical shift perturbation. Residues at the distal and proximal end (residues 1–5 and 33–37) exhibit relatively lower perturbation. These perturbations arise both due to direct contacts with the ligand as well as conformational changes occurring in the N-terminal region upon ligand binding. Overall, we found a good agreement between the residues predicted in this work from simulations to have a large chemical shift perturbation and those reported earlier using NMR (see S7 Fig). These residues are pictorially depicted in Fig 5C. The residues highlighted in cyan were predicted by simulations to have a large chemical shift and residues in orange and yellow have been identified in previous experiments [37,42]. We observe a considerable overlap in these residues, although many more residues were predicted to have a large chemical shift perturbation from our simulations relative to those identified using NMR. Nonetheless, a remarkable consistency is observed in the chemical shift perturbations predicted from coarse-grain simulations and those determined from NMR studies.

Interestingly, the chemical shift perturbations do not exactly match the interactions identified between the CXCR1 N-terminal region and the ligand from our simulations. In particular, a comparison of Fig 5A and 5B shows that residues 20–25 have a high contact probability, but low chemical shift perturbations. Similarly, residues 17–20 exhibit higher chemical shift difference relative to the corresponding contact probability. It is apparent that these chemical shift perturbations include environment effects due to altered conformational dynamics of the N-terminal region, particularly due to the C-shaped conformer adopted in the ligand-bound form. Since chemical shift perturbations are often used as a direct reporter of protein-protein contacts, we propose that caution should be exercised while interpreting such data, especially for intrinsically disordered regions. We believe that a combined approach integrating NMR and MD simulation approaches could provide novel insight into functional GPCR-ligand dynamics.

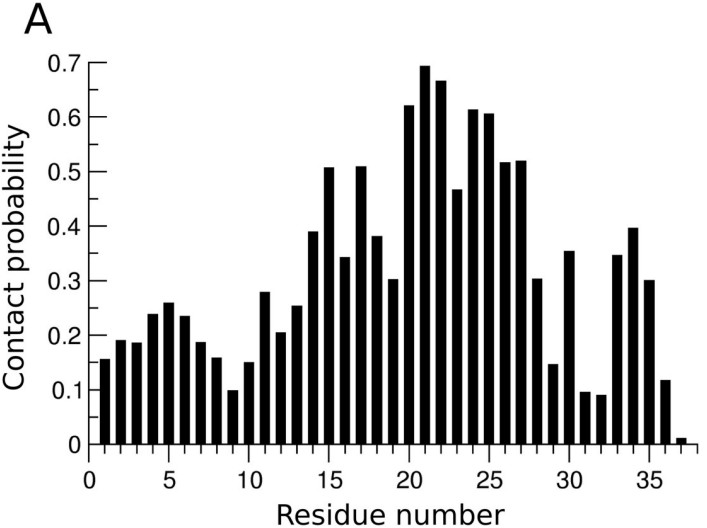

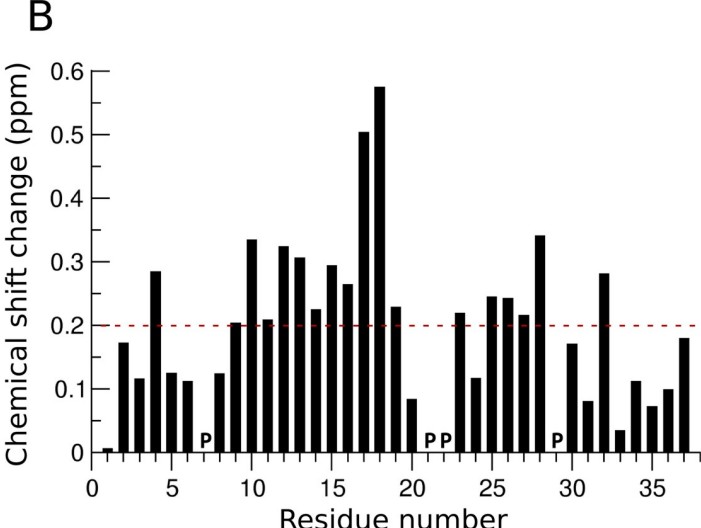

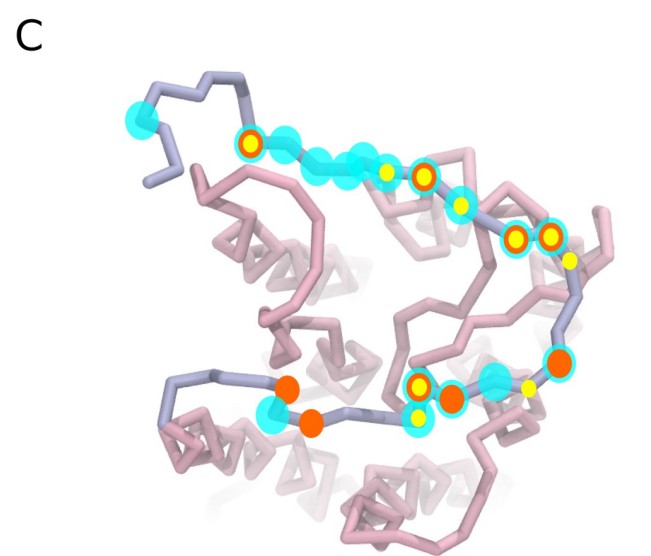

**Fig 5. Residue-wise interactions of the N-terminal region of CXCR1 with IL8.** (A) Residue-wise contact probabilities of the N-terminal region interacting with IL8. (B) Predicted chemical shift changes in the N-terminal region between the *apo-* and ligand-bound state. (C) The N-terminal residues with chemical shift perturbations above a cutoff (dotted lines in panel (B)) mapped onto the receptor structure. The cyan transparent spheres represent residues from the predictions. The orange and yellow spheres represent residues showing significant chemical shift changes as reported from NMR measurements [37,42]. See Methods and text for more details.

## Dynamic protein interactions define the chemokine N-domain and receptor interface

The dynamic interactions reflected in the contact probabilities at the CXCR1 N-terminal region (see Fig 5A) were observed in the ligand as well. We clustered the conformers corresponding to the different binding modes of IL8 with the CXCR1 N-terminal region. The five clusters that were observed to be most populated are shown schematically in Figs 6A and S8. Overall, it appears that the receptor N-terminal wraps around the ligand (IL8) and explores several binding modes. The main binding mode (~40% population) indicates that maximal interactions are localized with the N-domain and α-helix of IL8. The second and third binding mode additionally involves β1 and β3 strands, respectively. Residues involved in maximal contact of IL8 with the N-terminal region of CXCR1 were identified and mapped onto the structure, along with residues reported from NMR [37,42] and mutagenesis experiments [43–47] (Fig 6B). As expected, residues from the N-domain and α-helix were found to be involved, together with residues from the β1 and β3 strands, in IL8-CXCR1 N-terminal domain interaction. Importantly, we found an overlap between the regions in IL8 predicted to interact with the receptor and those reported previously. However, the N-terminal residues predicted to be important from mutagenesis studies [43–47] were not observed in our simulations or NMR studies [37,42]. The conformational plasticity of CXCR1 N-terminal region and dynamic interfaces sampled in the protein-protein complex appear to be a hallmark of chemokine-receptor binding.

## Discussion

The chemokine family of receptors are an important class of GPCRs that bind to the chemokine signaling proteins *via* their extracellular domains with a partial involvement of the transmembrane helices [23]. A molecular resolution of CXCR1-IL8 interactions would open up avenues for therapeutic design and an overall understanding of immune signaling. In this work, we have addressed the molecular details underlying chemokine-receptor interactions focusing on the representative pair, CXCR1-IL8. In particular, we have analyzed the structural dynamics of the N-terminal region of CXCR1 in both *apo-* and ligand-bound forms. In the *apo-*receptor, the N-terminal region is highly dynamic, consistent with the absence of resolution by NMR [34] and in agreement with its intrinsically disordered nature [38,39]. Upon ligand binding, the N-terminus adopts a dynamic C-shaped conformation that facilitates ligand binding *via* an extensive and dynamic surface. Our results are in overall agreement with chemical shift differences reported from NMR studies. Taken together, our results represent an important step toward understanding chemokine-receptor interactions, especially with respect to the first site of binding.

An important finding from our work is the inherent conformational dynamics of the N-terminal region and the binding interface. The identification and prediction of molecular details underlying such protein-protein interfaces is challenging in the context of GPCR-ligand interactions. In mechanistic terms, the main challenges are (i) resolving distinct temporal/spatial interactions (two-site/two-step model), (ii) accounting for the dynamics of the intrinsically

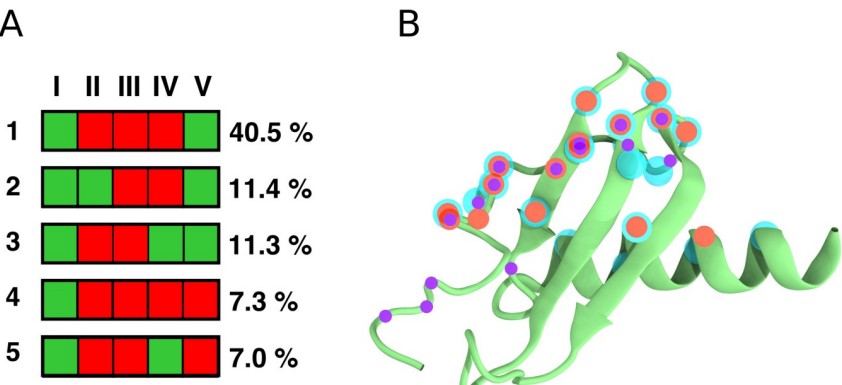

**Fig 6. Binding modes of IL8 characterizing its interactions with the N-terminal region of CXCR1.** (A) The most populated binding modes of IL8 characterized by the contacts formed by each of its structural element with the N-terminal region of CXCR1. The structural elements are denoted as I: N-domain, II: β1-strand, III: β2-strand, IV: β3-strand, and V: α-helix. The binding modes are numbered 1 to 5, in decreasing order of population. The green and red boxes represent interacting and non-interacting regions, respectively. (B) IL8 residues involved in binding to CXCR1 mapped on the cartoon representation of IL8. The cyan spheres represent interacting residues identified from our simulations. The orange and violet spheres represent interacting residues determined from previous NMR [37,42] and mutagenesis [43–47] studies, respectively. See Methods and text for more details.

disordered N-terminal region, and (iii) inherent technical difficulties in resolving the structural dynamics of membrane receptors. We observed differential conformational dynamics sampled by the N-terminal region in the presence and absence of the ligand. Interestingly, the *apo*-receptor samples a sub-space overlapping with the IL8-bound N-terminal region dynamics (Fig 4), suggesting a conformational selection by the ligand in the *apo*-receptor. Counterintuitively, the larger dynamics in the *apo*-receptor is associated with increased intra-protein contacts, whereas the C-shaped ligand-bound complex exhibits reduced intra-protein contacts. These loss of contacts within the N-terminal region in the IL8-bound complex are replaced by ligand contacts in the dynamic ligand-receptor interface. The dynamic protein-protein interface observed here represents an important aspect in the emerging understanding of plasticity in GPCR complexes [48].

We observe that the N-terminal region is the first site of ligand binding in the CXCR1 receptor, consistent with models based on previous fluorescence and NMR studies [36,37]. In the simulations reported here, the chemokine adopts a peripheral arrangement and a deeper binding of N-domain in the receptor lumen was not observed. This mode of binding differs from crystal structures of other chemokine receptors, but is consistent with CXCR1 NMR data [37]. In addition, a recent cryo-electron microscopy (cryo-EM) structure of a ternary complex of CXCR2, IL8 and G-protein reports that IL8 displayed a shallow binding mode compared to the other co-crystal structures of chemokines and their receptors [49]. The extensive contact surface between the ligand and the receptor N-terminal region are consistent with recent hypothesis from experimental approaches in related receptors [50]. In this work, we have compared chemical shift perturbations predicted from our simulations with results from NMR studies. Although the overall trends match quite well, we believe that the differences in the quantitative values could arise from the differential ensemble averages of experiments and simulations (due to different time scales associated with these approaches), peptide constructs used in experiments, and inaccuracies in prediction tools. Interestingly, our results indicate that the residues with maximum interactions do not necessarily exhibit the highest chemical shift perturbation. In the case of intrinsically disordered regions, there may not be a direct correlation between residues with high chemical shift perturbations and those at the interface of

the receptor-ligand contacts [51]. Instead, altered conformational dynamics of receptor N-terminal region (as reported here) could influence the observed chemical shift perturbations.

Computational studies, in close link with experimental approaches, have attempted to overcome some of the resolution problems associated with structure-based experiments. Several studies have combined docking followed by short MD simulations [52,53] and have been able to capture important interactions, such as electrostatic interactions at site-I. Computational design of chemokine binding proteins, such as receptor-derived peptide capture agents from the extracellular domains of CXCR1 [53] has also been reported. Similar approaches combining docking with free energy calculations were used to design IL8-based peptide inhibitors to inhibit binding of CXCR1 [54]. To circumvent the problem of limited sampling, coarse-grain simulations coupled with replica exchange have been successfully used for predicting conformational ensembles associated with the binding of a cyclic peptide antagonist to CXCR4 [55]. Coarse-grain simulations, in particular, appear to be well suited to predict protein-protein interactions within the membrane, such as in single transmembrane helical receptors [56,57] and GPCRs [58–61].

An emerging theme from the current work is its general relevance to intrinsically disordered domains, especially in the context of therapeutic design. Restriction of conformational flexibility upon protein binding by a "coupled folding and binding" model has been suggested to be a common mechanism [62], although specific examples are limited. For a more detailed understanding of the conformational space available in the presence and absence of protein partners, a Markov state model based analysis would be needed for rigorously identifying and estimating stationary populations of key macrostates, binding constants, (on and off) rate constants and the pathways of association. Although this has been previously reported for soluble protein-ligand complexes [63], it remains challenging for protein-protein complexes involving intrinsically disordered domains. An important point to consider would be the dimerization of the receptor and the ligand, which could alter ligand binding. However, the 1:1 stoichiometry (1 receptor:1 ligand) has been reported to probably be the predominant signaling form [24–26]. Another limitation of the current work is the simple membrane model considered. However, several of the experimental studies (fluorescence, NMR) have been performed in model membranes or even detergent micelles. The main salient features of the interaction are not suggested to differ, although the binding would be modulated overall.

In conclusion, we have used a combined atomistic and coarse-grain simulation approach to analyze the mechanism of binding of the chemokine IL8 to its cognate receptor CXCR1. We were able to observe the dynamic interfaces formed during the binding of CXCR1 and IL8. In addition, our results show that a conformational restriction of the flexible N-terminal region of the receptor induced by the ligand governs chemokine binding. These results suggest a conformational selection by the chemokine during the binding. The complementarity in shape and dynamic protein-protein interface appears to drive chemokine recognition by the receptor. We believe that our results represent an important step toward robust analysis of complex GPCR-ligand interactions and in designing improved therapeutics.

## Methods

### System setup and simulation parameters

The sequence of human CXCR1 N-terminal region (residues 1–37) was taken from the UniProtKB database (ID: P25024) and the structure was modeled in an extended conformation using Discovery Studio 3.5 (Accelrys Software Inc., Release 3.5, San Diego, CA). The *apo*-CXCR1 structure considered in this study was built by coupling the modeled structure of N-terminal domain to the NMR structure of CXCR1 (PDB ID 2LNL: residues 38–324). The

energy of this modeled structure was minimized (50,000 steps) using the steepest descent method. The structure was then mapped to its coarse-grain representation using parameters from the Martini v2.1 force field [64,65]. The receptor was embedded in a pre-equilibrated 1-palmitoyl-2-oleoyl-*sn*-glycero-3-phosphocholine (POPC) bilayer (284 lipids) using insane. py script [66] and then solvated. Twenty replicate simulations of 10 μs each were carried out for *apo*-CXCR1. The conformations of the N-terminal region sampled during these simulations were clustered, and two distinct receptor conformations were chosen, one with the N-terminal region coiled on the top of the receptor (receptor-contacted) and other with the N-terminal region interacting with the membrane bilayer (membrane-bound). For the ligand binding simulations of the two conformers (receptor-contacted and membrane-bound), IL8 was inserted at a distance of ~3 nm from the receptor to avoid potential bias arising from pre-placement. We considered two different orientations of IL8 while building these setups, resulting in four unique starting configurations of the CXCR1-IL8 simulations. The coarse-grain representation of IL8 was obtained by mapping from the atomistic three-dimensional structure (PDB ID: 1ILQ). Forty simulations of 10 μs each were run from these starting structures, both with and without elastic potential functions to fix the structural domains in IL8 [67]. The remaining parameters and setup were same as that of the CXCR1-IL8 system. The total simulation time was 400 μs, corresponding to 1.6 ms of atomistic sampling time.

All simulations were performed using the GROMACS-4.5.5 package [68,69]. For coarse-grain simulations, Martini force field (versions 2.0 and 2.2) [64,65] was used to represent lipids and proteins, respectively. Standard parameters corresponding to the coarse-grain Martini simulations were used. Non-bonded interactions were modeled using a cutoff of 1.2 nm. Electrostatic interactions were shifted to zero in the range 0 to 1.2, whereas Lennard-Jones potential was shifted to zero in the range of 0.9 to 1.2. Temperature was coupled to a thermostat at 300 K with a coupling constant of 0.1 ps using the v-rescale thermostat [70]. Pressure was coupled at 1 bar with a coupling constant of 0.5 ps using the semi-isotropic Berendsen algorithm [71] independently in the plane of the bilayer and perpendicular to the bilayer. Production runs were performed with a time step of 20 fs. Initial velocities for the systems were randomly chosen from a Maxwell distribution at 300 K.

The atomistic model of *apo*-CXCR1 was used as a starting structure for the all-atom MD simulations with CHARMM36 force-field parameters [72,73]. The receptor was inserted in a pre-equilibrated POPC bilayer using the CHARMM-GUI module [74]. Water and chloride ions were added to solvate and neutralize the charge on the system. Energy minimization was performed to remove steric clashes. The system was equilibrated under NVT conditions for 100 ps, followed by NPT ensemble for 1 ns, with position restraints on the receptor backbone. Atomistic simulations (1 long simulation of 1 μs and 6 sets of 100 ns each from different conformers) totaling to 1.6 μs were carried out as a control set. In the atomistic simulations, temperature coupling was applied with the v-rescale thermostat [70] to maintain temperature at 300 K. Semi-isotropic pressure coupling was applied to maintain a pressure of 1 bar along the direction of bilayer plane and perpendicular, using a Parrinello-Rahman barostat [75]. The long-range electrostatic interactions were treated with the particle mesh Ewald (PME) algorithm. The short-range electrostatic interactions and Lennard-Jones interactions were cutoff at 1.2 nm. A time step of 2 fs was considered for atomistic simulations.

## Analysis

Simulations were analyzed using in-house scripts, VMD [76] and GROMACS utilities. The residue-wise contacts were calculated using the g_distMat tool (*https://github.com/rjdkmr/g_distMat*). For a given pair of residues, a contact was defined if the minimum distance between

the residues (distance of closest approach) was within the cutoff (0.6 nm). The contact probability was calculated for each residue pair as the time for which they were in contact, normalized over the simulation length and averaged across all the simulation replicates.

A reference structure was identified from clustering the conformations by pooling all simulation sets in order to project the entire phase space sampled with respect to the RMSD to this reference structure. The clustering was performed using the GROMACS utility gmx cluster using the original GROMOS algorithm implemented in it [77]. The RMSD of the N-terminal region was used to cluster the conformers. The reference structure corresponds to the average structure of highest population cluster I. To analyze the conformational landscape, we projected the population densities along two vectors: RMSD to this reference structure and the distance between residues 1–26. The populations projected on the 2D surface were binned (0.1 nm) and averaged.

To calculate chemical shift changes in the CXCR1 N-terminal region upon IL8 binding, we considered the main structures sampled and a single conformer from each set was chosen from the highest population cluster. These conformers were transformed to the atomistic description (CHARMM36 force field) using Martini analysis tools [78]. The mapped structures were further equilibrated through energy minimization and short molecular dynamics simulation runs. These structures were provided as an input to the SHIFTX2 program [79] which predicts chemical shifts of backbone amides. The chemical shift values were averaged over replicates and chemical shift changes were calculated using the equation:

$$\Delta\delta = \sqrt{\frac{\left(\Delta\delta_H\right)^2 + \left(\frac{\Delta\delta_N}{5}\right)^2}{2}} \tag{1}$$

where $\Delta\delta_H$ is the change in the backbone amide proton chemical shift and $\Delta\delta_N$ is the change in the backbone amide nitrogen chemical shift.

## Supporting information

**S1 Fig. Snapshots of three-dimensional structures of CXCR1 and IL8.** NMR structure of (A) CXCR1 (PDB ID: 2LNL) with unresolved region of the N-terminus highlighted as a gray tube and (B) interleukin-8 (PDB ID: 1IL8). Extracellular domains of CXCR1 *viz*. ECL1, ECL2, ECL3 and N-terminus are colored as green, light blue, pink and red, respectively. IL8 is shown in cyan, with each region labeled.
(TIF)

**S2 Fig. Conformational dynamics of the N-terminal region of CXCR1 from all-atom and coarse-grain simulations.** (A) Residue-wise contact probabilities of the N-terminal region are plotted for *apo*-CXCR1 in coarse-grain simulations (upper diagonal) and atomistic simulations (lower diagonal), averaged over all simulation sets. The color bar displays probability of interactions between each residue-pair. (B) A plot of secondary structures of the N-terminal residues along the atomistic simulation trajectory. The secondary structure was calculated according to the DSSP algorithm [1]. White, red, yellow, black, green, blue and gray stretches represent coil, β-sheet, turn, β-bridge, bend, α-helix and $3_{10}$-helix, respectively.
(PDF)

**S3 Fig. Binding of IL8 to CXCR1 monitored over time.** Minimum distances between IL8 and CXCR1 are plotted as a function of time for forty simulations. The white and blue stretches represent unbound and ligand-bound regimes, respectively.
(TIF)

**S4 Fig. Key residue-residue interactions within the N-terminal region of CXCR1.** Normalized population histograms of distances between center of masses of side chains of residue pairs (A) Met1(yellow)-Asp26(red), (B) Thr5(green)-Thr18(magenta), (C) Phe17(gray)-Tyr27 (blue), (D) Ser2(violet)-Thr18(magenta), (E) Phe12(pink)-Pro29(maroon) and (F) Met1(yellow)-Asp13(orange). The black lines represent *apo*-CXCR1 and red lines represent CXCR1-IL8 simulations. Representative top-view snapshots from *apo*-CXCR1 (left) and IL8-bound (right) CXCR1 N-termini are displayed on top of each histogram. The N-terminal region is shown in cyan and the rest of the CXCR1 receptor is in pink.
(TIF)

**S5 Fig. Representative conformers of the N-terminal region of CXCR1.** The most populated conformations (side view) are shown for the *apo-* (left) and ligand-bound (right) forms of the receptor. The N-terminal region is shown in cyan and the rest of the receptor is in pink. The phospholipid headgroups are represented as yellow beads and acyl chains are in gray.
(TIF)

**S6 Fig. Interactions of the N-terminal region with the receptor.** Contact maps of the N-terminal region in the C-shaped conformation interacting with the extracellular domains of the receptor for (A) *apo*-CXCR1 and (B) IL8-bound CXCR1.
(TIF)

**S7 Fig. Chemical shift perturbations induced in the N-terminal region interacting with IL8.** (A) Predicted and (B) experimental chemical shift changes in the N-terminal region between the *apo-* and ligand-bound states. (C) Chemical shift differences plotted as a line graph for experimental (black) and predicted (red) values.
(TIF)

**S8 Fig. Binding modes of CXCR1-IL8 interactions.** The five dominant binding modes (1–5) are shown. The receptor is shown in pink, the N-terminal region is shown in blue, and the IL8 is represented in green.
(TIF)

**S1 Data. Details of system setup and input files for *apo-* and ligand-bound CXCR1 conformations.**
(ZIP)

**S2 Data. Coordinates of CXCR1-IL8 binding modes.**
(ZIP)

## Acknowledgments

We gratefully acknowledge computing resources from CSIR-NCL, CSIR-Fourth Paradigm Institute and PARAM Brahma Facility under the National Supercomputing Mission (Govt. of India) at the Indian Institute of Science Education and Research Pune. A.C. thanks Sreetama Pal for help and discussion during the preparation of the manuscript. We thank members of the Chattopadhyay laboratory for critically reading the manuscript and for their comments.

## Author Contributions

**Conceptualization:** Manali Joshi, Amitabha Chattopadhyay, Durba Sengupta.

**Formal analysis:** Shalmali Kharche, Durba Sengupta.

**Funding acquisition:** Amitabha Chattopadhyay, Durba Sengupta.

**Investigation:** Shalmali Kharche.

**Methodology:** Shalmali Kharche, Durba Sengupta.

**Resources:** Amitabha Chattopadhyay, Durba Sengupta.

**Supervision:** Manali Joshi, Durba Sengupta.

**Writing – original draft:** Shalmali Kharche, Durba Sengupta.

**Writing – review & editing:** Amitabha Chattopadhyay, Durba Sengupta.

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
