## [Decision Letter · Decision Letter 0]

26 Jan 2021

Dear Dr. CHATTOPADHYAY,

Thank you very much for submitting your manuscript "Conformational plasticity and dynamic interactions of the N-terminal domain of the chemokine receptor CXCR1" for consideration at PLOS Computational Biology.

As with all papers reviewed by the journal, your manuscript was reviewed by members of the editorial board and by several independent reviewers. In light of the reviews (below this email), we would like to invite the resubmission of a significantly-revised version that takes into account the reviewers' comments.

We cannot make any decision about publication until we have seen the revised manuscript and your response to the reviewers' comments. Your revised manuscript is also likely to be sent to reviewers for further evaluation.

Sincerely,

Turkan Haliloglu

Associate Editor

PLOS Computational Biology

Nir Ben-Tal

Deputy Editor

PLOS Computational Biology

A suggestion by Nir: The authors may want to correlate their simulations with evolutionary data (e.g., using ConSurf), where the premise is that biologically important sites that are mechanistically critical would often be highly conserved. It may further consolidate the simulations. But please feel free to ignore. 

Reviewer's Responses to Questions

**Comments to the Authors:**

Reviewer #1: The article “Conformational plasticity and dynamic interactions of the N-terminal domain of the chemokine receptor CXCR1” by Shalmali Kharche, Manali Joshi, Amitabha Chattopadhyay, Durba Sengupta presents a study of the interaction between the CXC chemokine receptor 1 and interleukin-8 by molecular dynamics.

This is an important subject, well introduced, and the results are generally very convincing. The main results are: 1) the dynamics of the N-terminal region undergo a conformational selection upon ligand binding, i.e. it switches from being unstructured in the apo-receptor to a C-shaped conformation upon ligand binding. 2) The comparison with NMR data shows that the results are generally supported by experiment. Interestingly the authors noticed that the chemical shifts are at least partially due to the change in conformation, calling for caution in the interpretation of chemical shift experiments. This second result is interesting per se.

The article would benefit, however, from considering a few points:

Major points

1) One limitation is that the authors use clustering but neither detail the underlying methods nor present the clustering itself. It would be useful for the reader to see the clustering, even as supplementary material. One potential question is: How many clusters should be considered?

In principle, the authors could even consider PCA in terms of analysis of their trajectories, given that they appear to be converged.

2) It would be useful to see the different binding modes (Fig 6). It would also be good to know how many times each one occurred. Does it depend on the starting orientation? Incidentally it is unclear whether, when binding occurred more than once, whether rebinding occurred with the same binding mode. Finally, one may wonder whether the data presented in figure 2c depends on the binding mode.

3) It is not completely clear how the contacts presented in Fig 3 were selected. Why not mention residue 10 and its interactions with residues 17, 20 and 29? or the interaction between 12 and 17? The interaction between 12 and 29 is also ignored whereas it is presented in Fig S3. This is important because the interaction between residues 1 and 26 is then used as a collective variable and one may wonder whether the picture given by fig 4 could be different if one other residue pair would have been considered.

4) The comparison with experimental data is an important aspect of the study and should be made as quantitative as possible. If the authors cannot get the numbers from the authors of the NMR experiments perhaps they could make the comparison more stringent by playing on the threshold, in order to reduce the number of residues with high computed chemical shift.

Minor points:

5) For the conformational landscape presented in figure 4 it would be interesting to have more information about the clustering. It would also be interesting to have information about the dynamics, i.e. do the authors observe fluctuations between the basins during the simulations. As for the structures shown, it would be interesting to have also a view in the membrane plane.

6) The receptor is known, as mentioned by the authors, to form dimers. It would be interesting for the reader to have an idea of whether the results could be altered in the dimer. For instance, the authors could discuss/comment on the potential overlap between the dimerization interface and the interface with IL8 or area explored by the Nter.

7) Given that interaction with the membrane is known to affect the function of the receptor, and the nter to interact with the membrane, it would be worth commenting/discussing on the potential impact of having the membrane represented by a pure POPC bilayer.

Reviewer #2: The conformational dynamics of N-termindal domain of CXCR1 with interleukin-8 (IL-8) has been explored using state-of-the-art computer simulation approaches by Kharche and coworkers. The authors first explored the dynamics of apo-protein using coarse-grained simulation and then validated its prediction against all-atom simulation. Subsequently, the model investigated the protein-protein association dynamics using coarse-grained simulation for extensive period of time and pin-pointed the key locations of the association process. The manuscript is well-written and most of the analysis is clear. The reviewer has enjoyed reading the draft and found it commendable to explore recognition process via coarse-grained simulation, an emerging approach. The referee believe that this manuscript is well-suited for publication in PLOS com. biol. . However, the author should explore following suggestions of the referee while revising the manuscript:

1. ‘The final bound complex’ is a vague term in this work. In typical protein-ligand binding simulation, generally the ‘binding simulation’ are terminated after the bound pose agrees with the crystallographic pose. Here, In absence of experimentally known pose, the criteria for termination of this long simulation should be properly justified.

2. The author’s claim that the the dynamic complex sampled by the simulation is validated by NMR data is a bit of stretch. The question is , in absence of a crystallographic or cryo-EM data, the reviewer is not very convinced how a dynamic complex’s bound location can be validated by an average quantity like chemical shift. That might be the reason for apparent difference between predicted and experimental chemical shift prediction. The authors should discuss the reason for the discrepancy in a more clear way.

3. In the related context, the author mention that the ‘to compare this data with simulations reported here, we chose representative structures from each of the coarse-grained simulation and mapped them to their atomistic representation’. The authors should clarify how the representative structure was chosen and how they mapped it to all-atom data. In the method section, the authors have cited reference 64 but that is a paper on insane tool. Is this what they meant? Or is it the back-mapping paper by Marrink and coworkers they used? Nonetheless, more details should be provided.

4. The authors should provide a snapshot highlighting ECL1, ECL2, ECL3, Lumen etc in the protein. Otherwise, it is not clear which locations are they referring to. On a related point, the authors should provide a 3D spatial density maps of the encounter of IL8 around protein.

5. Page 12: The choice of collective variable. The referee understands the reason behind choice of first one i.e backbone RMSD of N-terminal domain. However, it is not Clear how they have chosen the second one. There should be more justification, even if it is merely via visual explanation.

6. The work report an impressive number of simulation trajectories. This sets an ideal ground for developing a Markov-state-model based analysis for rigorously identifying nd estimating stationary populations of key macro states, predicting eventual binding constants, (on and off) rate constants and the pathways. There are precedences of this . Recently, protein-ligand binding using coarse-grained model has been looked at within the framework of Markov state model. The reviewer understand that this is a formidable job and might itself be an independent work that the authors can seriously think about as a future work to make use of the current trajectories and at least should provide a discussion in the current manuscript.

Reviewer #3: Kharche et al. has presented a comprehensive simulation-based approach to map the conformational and binding landscape of CXCR1. Even though the approach includes extensive simulations, it lacks the following critical points, which have to be addressed for a publication in Plos Comp Bio.

Abstract:

** The abstract is written in an unclear fashion. The following should be addressed to make it direct and concise.

- Line #35: The sentence starting with "Although .." is too long.

- Line #39: What do you mean by "validated atomistic models"? This is unclear, please clarify. Also, you cannot validate your results with NMR Chem Shift predictions, which you derived from your simulations.

- Line #41: The validation by NMR does not include a 100% agreement with the exp and comp data. Please explicitly indicate this in the abstract.

- Line #43: The authors did not present any data to claim that their simulation data is more reliable than the NMR data. So, how could they claim that NMR data should be used with caution.

Introduction:

** Line #72: The sentence starting with "The high flexibility .." is not clear, please rephrase.

** The binding models, the domain organizations & available structures of CXCR1 and IL8 should be illustrated with a figure.

Results:

** Line #414: What are the numbers to come up with the conclusion that "N-terminal relaxed quickly ...". Please quantify.

** What is the population percentages of the two main conformers observed (membrane-bound and receptor-contacted)? How close are the conformations within a single group? Please quantify.

** Rg is not very sensitive to group different conformations. Therefore, the authors should present other metrics to convince the reader about the dominant populations observed.

** Line #178: What is the length of the all atom simulation? Did you perform only one simulation? Note this also in the results.

** The authors did not provide any quantitative mean to compare the conformations sampled in the atomistic and coarse-grain simulations. A visual contact map comparison would not be enough here.

** I am very skeptical about the approach the authors used to predict CXCR1-IL8 interactions. I am really curious why the authors did not follow the following and tried to predict CXCR1-IL8 interactions with classical MD (which does not seem to fit to their purpose):

- Isolate the most dominant apo conformers from MD. Gather the available mutagenesis and NMR chem shift data. Use the apo CXCR1-IL8 ensemble, together with the experimental data to predict the binding of CXCR1-IL8 with information-driven docking. Among the available methods, HADDOCK, for example, is perfectly capable of doing this. The authors can then simulate the most viable docking models with MD to analyze their interfaces.

** Minimum distance between two monomers, as presented in Fig 2a, cannot be a reporter of specific interaction. Please analize the established interfaces in more detail to deduce solid data on binding.

** For Figure 3a: Again here, we need a quantitative comparison.

** Figure 4a: The authors tried to cluster the conformers based on one single distance. They should use different measures (inter-monomer contacts, for example) to see whether their clustering holds. Clustering based on one distance could be misleading.

** The authors should present the available experimental data. I could not find their explicit description. For example the agreement between computed and measured NMR data should be number-wise presented in Fig5. Without showing this, it would be bold to claim that simulation data supersedes experimental data and therefore caution should be taken while using experimental data.

Discussion:

** Should be updated according to the new findings.

Methods:

** What is the force field used during atomistic simulation? Also, how many simulations were performed here? If the authors performed one simulation, then they should run more.

** Why did the authors use a such an old version of GROMACS?

Reviewer #4: First of all, my apologies to the authors and editor for the delay in getting this review to you.

This manuscript uses extensive CG (and some atomistic) simulations to study the interactions of the key chemokine receptor CXCR1 with its ligand IL8, which serves to regulate innate immune responses.

The N-terminal region of the receptor is thought to bind to the ligand (similarly to other GPCRs), but this is currently structurally unresolved. Thus, CG simulations were used to predict possible N-terminal conformations in the context of a membrane environment and ligand bound states.

The key results are that IL8 stabilises the otherwise highly flexible / disordered N-terminus. Results were validated to a certain extent by comparison with atomistic simulations and prior NMR chemical shift and mutagenesis studies.

Overall, the work will be of interest to the membrane receptor community, and might also be useful for future drug design efforts (though this isn’t discussed). I would suggest the following should be considered:

1) The key focus here is the N-terminal region of CXCR1, and all the results depend on how this was treated. The CG simulations were performed starting from an extended N-terminal state, which was mapped to CG using Martini 2.1. It’s good that the authors provide their input files, but for the casual reader, the authors should say more about how the structure was treated in the manuscript. E.g. just angles/dihedrals (if so, did that result in a uniform set of extended secondary structure parameters?), or was an elastic network also implemented?

2) The authors conclude that the N-terminus samples similar conformations in CG and atomistic simulations (Fig 1 vs Fig S1). However, I would like to see e.g. the minimum distances and radii of gyration for the atomistic resolution. If things are not reproduced across resolutions perfectly, that’s okay, but it would be useful to know for other researchers in the field. It would also be useful for other researchers to know if they could just use the Martini forcefield “out of the box” in future to model such N-terminal regions in GPCRs, or if more extensive refinement against atomistic data might first be required.

3) How much faith do the authors have in the CG simulations in correctly predicting conformational changes in the N-terminal domain upon ligand binding? CG snapshots were converted to atomistic resolution, and then used to compute chemical shifts for comparison with experiments. But it might be expected that the method of conversion to atomistic representation may well affect this calculation; what would happen if the CG snapshots were first refined (even for just a few nanoseconds) in atomistic simulations? The authors warn that caution should be exercised when interpreting chemical shift perturbation data, which I agree with, but not just because of the experimental limitations.

4) The authors point out that both receptors and chemokines have been shown to dimerize in vivo – it would be good to discuss how this might affect the observations reported here.

5) Likewise, as noted by the authors, the loops of GPCRs are sensitive to lipid types. Simulations reported here were of simple POPC lipid membranes – do the authors expect that more realistic complex lipid compositions might affect their observations?

6) Would the authors be able to comment on whether their derived conformations might be useful in drug design? And for that matter, it would be nice if the authors made some of their dominant ligand bound structures available to the community.

**Have all data underlying the figures and results presented in the manuscript been provided?**

Reviewer #1: Yes

Reviewer #2: Yes

Reviewer #3: None

Reviewer #4: Yes

PLOS authors have the option to publish the peer review history of their article (what does this mean?). If published, this will include your full peer review and any attached files.

Reviewer #1: **Yes: **Antoine Taly

Reviewer #2: No

Reviewer #3: No

Reviewer #4: No
---

## [Decision Letter · Decision Letter 1]

28 Apr 2021

Dear Dr. CHATTOPADHYAY,

We are pleased to inform you that your manuscript 'Conformational plasticity and dynamic interactions of the N-terminal domain of the chemokine receptor CXCR1' has been provisionally accepted for publication in PLOS Computational Biology.

Best regards,

Turkan Haliloglu

Associate Editor

PLOS Computational Biology

Nir Ben-Tal

Deputy Editor

PLOS Computational Biology

Reviewer's Responses to Questions

**Comments to the Authors:**

Reviewer #1: The reviewer would like to thank the authors for their very serious work on revising the manuscript and addressing all questions and remarks?

Reviewer #2: The authors have addressed all referee concerns. The updated manuscript is recommended for publication in its current state.

Reviewer #4: The authors have done a good job of addressing the reviewers' comments.

**Have the authors made all data and (if applicable) computational code underlying the findings in their manuscript fully available?**

Reviewer #1: Yes

Reviewer #2: Yes

Reviewer #4: Yes

PLOS authors have the option to publish the peer review history of their article (what does this mean?). If published, this will include your full peer review and any attached files.

Reviewer #1: **Yes: **Antoine Taly

Reviewer #2: No

Reviewer #4: No

---

## [Editor Report · Acceptance letter]

17 May 2021

PCOMPBIOL-D-20-02189R1 

Conformational plasticity and dynamic interactions of the N-terminal domain of the chemokine receptor CXCR1

Dear Dr CHATTOPADHYAY,

I am pleased to inform you that your manuscript has been formally accepted for publication in PLOS Computational Biology. Your manuscript is now with our production department and you will be notified of the publication date in due course.

With kind regards,

Zsofi Zombor
